

# Adsorption of thallium(I) on rutile nano-titanium dioxide and environmental implications

Weilong Zhang[1], Yang Wu[1], Jin Wang[1], Juan Liu[1], Haifeng Lu[2], Shuijing Zhai[3], Qiaohui Zhong[1,4], Siyu Liu[1], Wanying Zhong[1], Chunling Huang[1], Xiaoxiang Yu[1], Wenhui Zhang[1] and Yongheng Chen[1]

[1] Innovation Center and Key Laboratory of Water Quality and Conservation in the Pearl River Delta, Ministry of Education, School of Environmental Science and Engineering, Guangzhou University, Guangzhou, China
[2] Wuhan Digital Engineering Institute, Wuhan, China
[3] Key Laboratory of Humid Subtropical Eco-geographical Processes, Ministry of Education, College of Geography Science, Fujian Normal University, Fuzhou, China
[4] Guangzhou Institute of Geochemistry, Chinese Academy of Sciences, Guangzhou, China

Corresponding authors
Juan Liu, liujuan858585@163.com, liujuan@gzhu.edu.cn
Shuijing Zhai, S2008shuijing@163.com

## ABSTRACT

Rutile nano-titanium dioxide (RNTD) characterized by loose particles with diameter in 20–50 nm has a very large surface area for adsorption of Tl, a typical trace metal that has severe toxicity. The increasing application of RNTD and widespread discharge of Tl-bearing effluents from various industrial activities would increase the risk of their co-exposure in aquatic environments. The adsorption behavior of Tl(I) (a prevalent form of Tl in nature) on RNTD was studied as a function of solution pH, temperature, and ion strength. Adsorption isotherms, kinetics, and thermodynamics for Tl(I) were also investigated. The adsorption of Tl(I) on RNTD started at very low pH values and increased abruptly, then maintained at high level with increasing pH >9. Uptake of Tl(I) was very fast on RNTD in the first 15 min then slowed down. The adsorption of Tl(I) on RNTD was an exothermic process; and the adsorption isotherm of Tl(I) followed the Langmuir model, with the maximum adsorption amount of 51.2 mg/g at room temperature. The kinetics of Tl adsorption can be described by a pseudo-second-order equation. FT-IR spectroscopy revealed that -OH and -TiOO-H play an important role in the adsorption. All these results indicate that RNTD has a fast adsorption rate and excellent adsorption amount for Tl(I), which can thus alter the transport, bioavailability and fate of Tl(I) in aqueous environment.

# INTRODUCTION

Thallium is a heavy metal whose toxicity to mammals is second only to that of methylmercury and far exceeds that of chromium, mercury, and lead (*Campanella et al., 2016*; *Campanella et al., 2017*; *Casiot et al., 2011*; *Galván-Arzate & Santamaría, 1998*; *Grösslová et al., 2018*; *Li, Xiao & Zheng, 2012*; *Liu et al., 2016a*; *Perotti et al., 2017*). Thallium can be accumulated in bone marrow, kidneys and different organs, thereby affecting the gastrointestinal and urinary tract, and even causing permanent damage in

muscle atrophy or central nervous system (*Galván-Arzate & Santamaría, 1998*; *Peter & Viraraghavan, 2005*). Hence, Tl was listed as one of the toxic pollutants of priority both in USA and China (*Xiao et al., 2012*).

Thallium pollution is mainly due to widespread use of Tl-containing minerals in industry. For example, mining of Hg-As-Tl deposits at Lanmuchang in Guizhou, southwestern China, led to severe Tl poisoning in local residents during the 1960–1970s (*Xiao et al., 2004a*; *Xiao et al., 2004b*; *Xiao, Guha & Boyle, 2004c*; *Xiao et al., 2007*; *Xiao et al., 2012*). Waste discharge from a Pb-Zn smelter using Tl-bearing minerals resulted in Tl pollution in the Northern Branch of the Pearl River in 2010 (*Liu et al., 2016b*; *Liu et al., 2017*).

In recent years, rutile nano-titanium dioxide (RNTD) have been widely employed in a variety of products (e.g., paints, cosmetics, optical component, biosensors and sunscreen) (*Kusior et al., 2017*). The rapid growth in $TiO_2$ production and its industrial applications may result in enhanced release into the environment and exposure to human (*Danielsson et al., 2018*; *Liu, 2005*). Having large surface area, $TiO_2$ nano-particles can alter the transport, bioavailability and fate of heavy metals by strong adsorption interactions. It is likely for Tl(I) and RNTD to encounter and co-exist in the aqueous environment. The adsorption of various heavy metals (such as Pb, Zn, Cd and Cu) on RNTD has been extensively investigated, but it remains unclear for the adsorption behavior of Tl on RNTD (Jiang et al., 2014).

In this study, batch experiments were conducted to explore the adsorption behavior of Tl on RNTD. The effects of various parameters—RNTD dosage, ionic strength, pH, time, initial concentration of Tl, and temperature—on the adsorption behavior, as well as the adsorption isotherms, kinetics, and thermodynamics of Tl(I), were investigated.

## MATERIALS AND METHODS

### Materials
Analytical reagent (AR)-grade RNTD was purchased from Aladdin Reagents Co., Ltd. (Shanghai, China). Thallium(I) nitrate (99.5%, metal basis) was purchased from Alfa Aesar, A Johnson Matthey Company. AR-grade $NaClO_4$, $HClO_4$, and NaOH were obtained from Guangzhou Chemical Reagent Co., Ltd. (China). All experiments were performed using Milli-Q water (Millipore Corp.).

### Batch adsorption experiments
All the batch adsorption experiments were conducted in a rotary shaker (25 ± 0.2 °C, 200 rpm) using a 50 mL PP centrifuge tube. The pH was adjusted to the desired value by using 0.1 mol/L $HClO_4$ or NaOH. Stock solution of $TlNO_3$ (100 mg/L) was prepared with deionized water.

First, the influence of RNTD dosage on the adsorption of Tl(I) was studied. For this purpose, the concentration of RNTD was set in the range 0.5 to 5 g/L at desired intervals, using 0.1 mol/L $NaClO_4$ as the background, to ensure constant ionic strength with pH 7.0 ± 0.3. To investigate the effect of ionic strength on Tl(I) adsorption, an appropriate amount of $NaClO_4$ was added with the concentration ranging from 0.05 to 3 mol/L at

desired intervals. The influence of pH on Tl(I) adsorption was studied in the range of 2 to 11. For isotherm studies, the temperature was set at 298 K, 303 K and 313 K, respectively, and the initial concentration of Tl(I) varied from 0.02 to 20 mg/L. The solutions obtained were filtered rapidly through a 0.45 mm membrane after 300 min, and Tl concentration was measured immediately by adsorption thermodynamics experiments. To evaluate the adsorption kinetics, samples were collected at desired intervals in the range of 0 to 300 min. The concentration of Tl(I) was determined by inductively coupled plasma–mass spectrometry (ICP-MS; NexION, PerkinElmer US).

The adsorption amount and adsorption rate are calculated as follows:

$$\text{Adsorption rate}: \alpha\% = \frac{C0 - Cs}{C0} \times 100\% \tag{1}$$

$$\text{Adsorption amount}: Qe = \frac{(C0 - Cs)V}{m} \tag{2}$$

Where, $C_0$—initial concentration (mg/L);

$C_s$—remaining concentration of Tl after adsorption (mg/L);

V—solution volume (L);

m—total mass of adsorbent added (g).

### Characterization

The morphologies of RNTD were observed by a JSM-7001F scanning electron microscopy (SEM, JOEL, Japan) integrated with an energy-dispersive X-ray spectrometer (Oxford Instruments, UK). To examine surface properties, the FTIR (https://www.sciencedirect.com/topics/chemistry/absorption-spectrum) of $TiO_2$ nanomaterials were analyzed by using a Bruker (TENSOR27, Germany) FTIR spectrometer in the frequency range of 400–4,000 $cm^{-1}$, operating with a spectral resolution of 2 $cm^{-1}$. The crystal phase of the sample was visualized by X-ray powder diffraction (XRD) using an X'Pert Pro Diffractometer (PANanalytical, EA Almelo, The Netherlands) X-ray diffractometry operating at 100 kV and 40 mA, using Cu Ka radiation at a scan speed of 4°/min.

## RESULTS AND DISCUSSION

### Effect of RNTD dosage

The adsorption effect of adsorbent on Tl is due to the existence of active adsorption sites in the adsorbent. Under a constant initial Tl concentration, increase of the dosage of adsorbent will enhance the contact site, thereby increasing the adsorption amount of adsorbent on Tl (*Shen et al., 2009*). However, a decrease in adsorption amount with increased dosage was observed (Fig. 1), and the adsorption rate increased from 55.4% to 92% as the dosage increased from 0.5 to 2.5 g/L. It did not continue to rise afterwards due to further addition of the adsorbent will cause difficulty in dispersion, which can hinder Tl adsorption. According to the literature, the mass of $TiO_2$ particles could increase the sedimentation aggregation (*He & Zhang, 2003*).
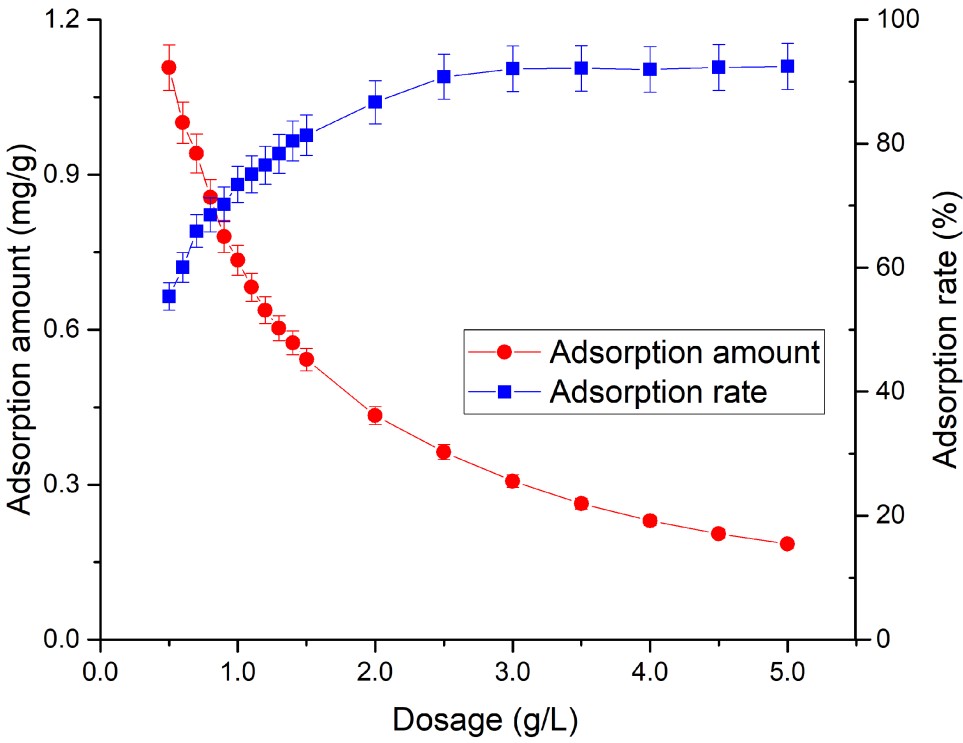

**Figure 1** **Effect of rutile nano-TiO₂ dosage on Tl adsorption.** The adsorption amount decreased with increased dosage, and the adsorption rate increased upmost to around 92%.

## Effect of initial thallium concentration

Initial concentration of Tl in natural systems will also influence Tl adsorption behavior (*Zhang et al., 2017*). As shown in Fig. 2, at 0.02 to 10 mg/L, as the initial concentration increases, the adsorption rate shows a tendency to decrease unevenly from 70%. From 10 to 20 mg/L, the adsorption rate no obvious change with the initial concentration. Obviously, without changing the amount of adsorbent, if initial Tl concentration in the water is higher, the nano TiO₂ can adsorb more Tl. The adsorption rate no longer decreases at initial Tl concentration of 10 mg/L. From the perspective of cost savings, the Tl concentration of 10 mg/L would be most suitable for adsorption by rutile nano TiO₂, which deserves highest environmental benefits.

## Effect of ionic strength

Another important factor affecting Tl(I) concentration in natural systems is ionic strength (*Liu et al., 2011*). As shown in Fig. 3, with an increase of $NaClO_4$ concentration, the adsorption rate gradually increased from 33.6% to 86.7% and then tended to reach equilibrium. The adsorption of Tl mainly depends on the surface negative charge of the adsorbent. Increasing $NaClO_4$ concentration is favorable for adsorption, but excessive concentrations reduce adsorption rate. When $NaClO_4$ concentration is >1.5 mol/L, the adsorption rate tends to become constant (*Vilar, Botelho & Rui, 2005*). At this time, the

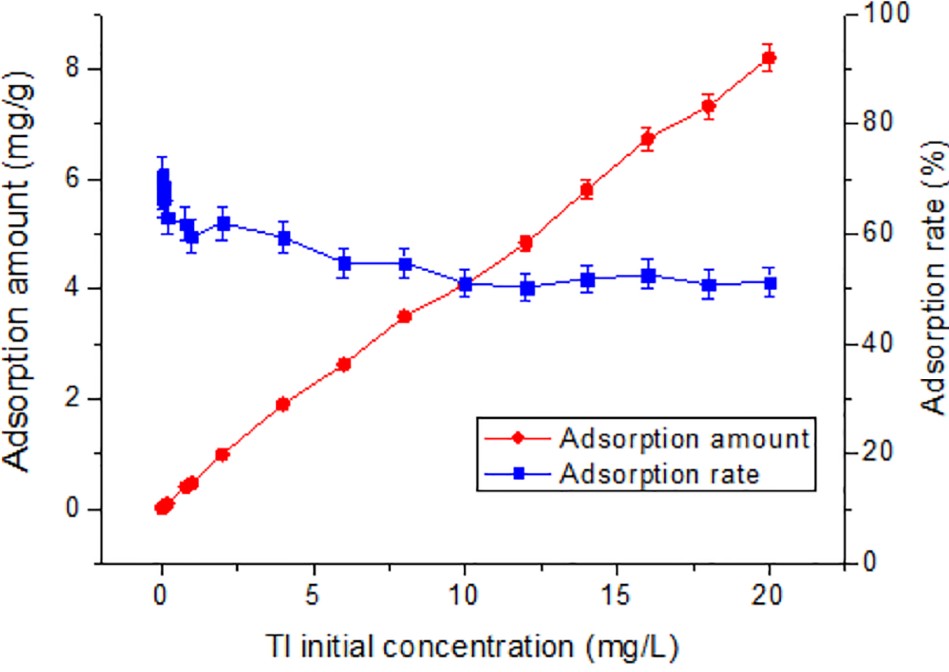

**Figure 2** **Effect of initial concentration on adsorption of Tl by rutile nano-TiO$_2$.** The adsorption amount increased with Tl initial concentration while the adsorption rate decreased slowly.

adsorption increases with the increase of the ionic strength or is not sensitive to the ion concentration. It is likely that the inner surface complex (ISSC) is formed between RNTD and Tl, the chemical bond between RNTD and Tl is generally a coordination covalent bond.

## Effect of pH

The effect of the pH value of the solution on the adsorption is mainly achieved by changing the charge carried by the adsorbent and the adsorbate, thereby affecting the electrostatic interaction between the adsorbent and the adsorbate (*Wu, Joo & Lee, 2005*; *Wu et al., 2004*). As shown in Fig. 4, the pH of the solution exerted a significant role on the adsorption of Tl(I). The adsorption percentage rised from 28.9% to 60.2% when the pH increased from 2 to 11. No obvious change in the adsorption occurred from pH 2 to pH 5. The adsorption amount increased slowly in the pH range from 2 to 5, but very significantly in the pH range from 5 to 9, followed by a plateau at pH 9 to 11. In aqueous solution, Tl$^+$ is the predominant species in the pH range of 2–11. The elevated adsorption amount with pH can be owing to pH$_{pzc}$ of the RNTD adsorbent (*Govender et al., 2007*; *Mahamuni et al., 1999*; *Vayssieres et al., 2001*). At pH values of 2∼5, which are lower than pH$_{pzc}$, the surface charge of the RNTD was positive. It facilitated great electrostatic repulsion between Tl$^+$ and positively-charged RNTD, which inhibited adsorption of Tl(I) ions. At pH values (6∼9) higher than pH$_{pzc}$, the surface charge of the TNTs turned negative, which promoted the adsorption of Tl$^+$, due to the electrostatic force of attraction between Tl(I) ions and the

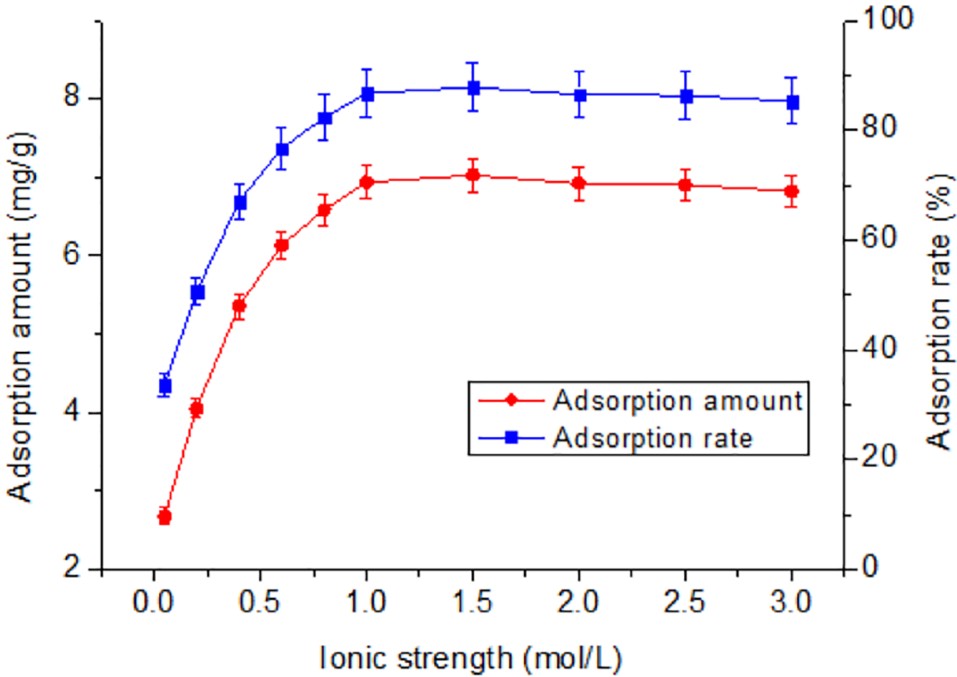

**Figure 3** **Effect of ionic strength on adsorption of Tl by rutile nano-TiO₂.** Adsorption amount and rate increased steadily and then levelled when the ionic strength was >1.5 mol/L.

surface. For further increase in pH, approximately 60% removal efficiency was achieved when pH was above 9, suggesting a fairy good adsorption performance of RNTD.

## Adsorption kinetics

As shown in Fig. 5, the adsorption amount rapidly increased from 2.8 to 3.3 mg/g in the first 5–15 min but did not change significantly further on, indicating a rapid adsorption of Tl(I) by RNTD. The adsorption process was complete within 15 min, The rapid uptake was mainly due to the large amount of active sites on the surface of the RNTD.

In order to better understand the behavior of Tl(I) adsorption and possible controlling mechanism, the kinetics experimental data were fitted with the commonly used kinetic models, such as the pseudo first-order, pseudo second-order, and Elovich models (*Liu et al., 2014*; *Pu et al., 2013*).

The pseudo-first-order kinetic model equation is expressed by:

$$\lg(Qe - Qt) = \lg Qe - (k1t)/2.303 \tag{3}$$

Where $Q_e$ is the adsorption amount (mg/g); $Q_t$ is the adsorption amount (mg/g) at time $t$; and $k_1$ is the first-order adsorption rate constant (g/mg min).

The pseudo-second-order kinetic model equation is expressed as follows:

$$t/Qt = 1/k2Qe^2 + t/Qe \tag{4}$$

Where $Q_e$ is the adsorption amount (mg/g); $Q_t$ is the adsorption amount (mg/g); $k_2$ is the secondary adsorption rate constant (g/mg min).

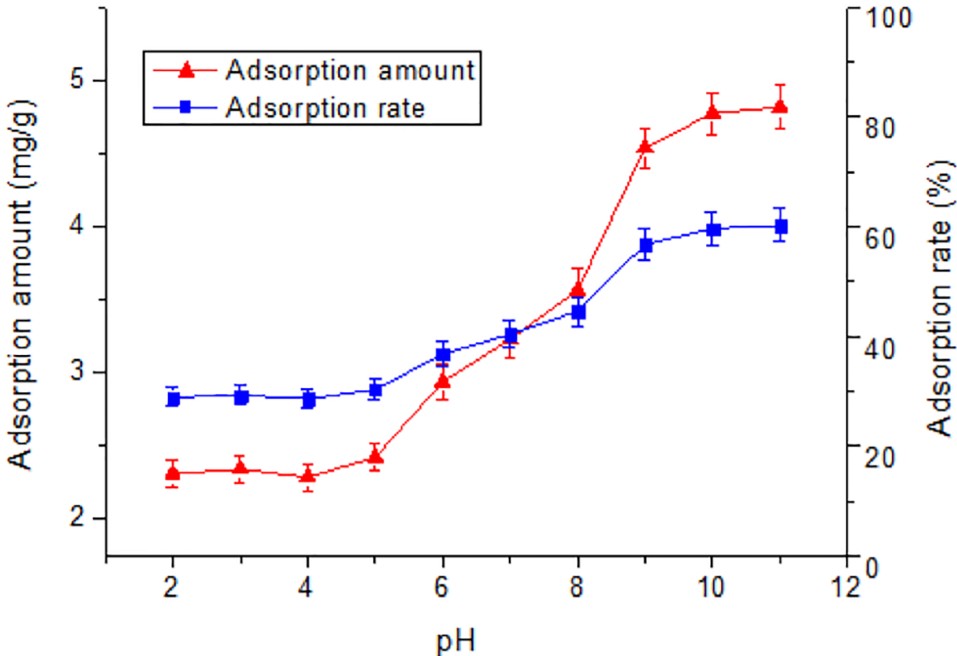

**Figure 4** **Effect of pH on adsorption of Tl by rutile nano-TiO₂.** The adsorption amount increased slowly in the pH range from 2 to 5, but very significantly in the pH range from 5 to 9, followed by a near plateau at pH 9 to 11.

**Table 1** **Kinetic parameters for adsorption of Tl(I) on rutile nano-TiO₂.**

| Quasi-first-order | | | Quasi-secondary | | | Elovich | | |
|---|---|---|---|---|---|---|---|---|
| $k_1$ | $q_e$ | $R^2$ | $k_2$ | qe | $R^2$ | $k_e$ | C | $R^2$ |
| 0.0014 | 3.0285 | 0.6169 | 0.0857 | 3.5157 | 0.9969 | 0.1229 | 2.7796 | 0.6947 |

Elovich dynamic model is expressed by:

$$Qt = ke \times \ln t + C \tag{5}$$

Where $Q_t$ is the adsorption amount (mg/g) at time t; $k_e$ and $C$ are constants; t is the adsorption time (min); and $k_e$ is related to the adsorption efficiency. The greater the $k_e$ value, the higher is the adsorption efficiency.

As listed in Table 1, the kinetic results are fitted by the pseudo-second-order model very well, with a high correlation coefficient ($R^2 = 0.9969$). This suggests that the rate-controlling step for the adsorption was the initial diffusion of metal ions from the solution and their subsequent interaction with the -ONa/-OH groups on the RNTD surface, and the subsequent interaction between the -ONa/-OH groups of RNTD and the metal ions.

### Adsorption isotherms

With an increase in the initial Tl(I) concentration, the adsorption rate showed a trend of decrease from 70% to 50% (Fig. 6). The lowest adsorption amount was found at the reaction temperature of 313 K, while higher adsorption amounts were observed at lower

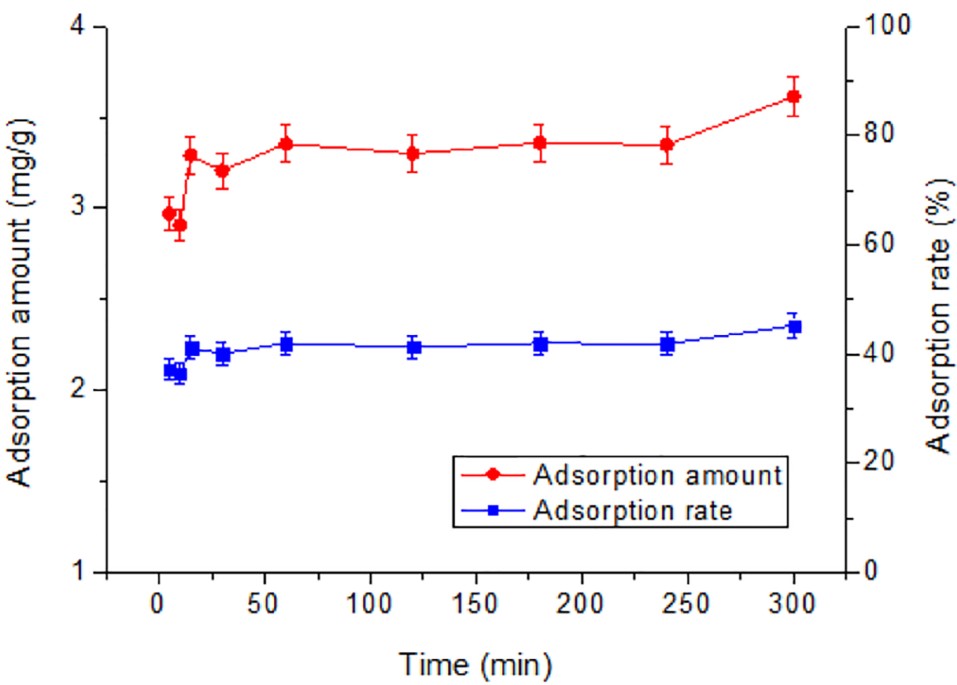

**Figure 5** **Effect of time on adsorption of Tl by rutile nano-TiO₂.** The adsorption process was rapid, and completed within 15 min.

temperatures, which indicates that the adsorption is exothermic. The adsorption amount of RNTD showed a near-linear dependent on the initial Tl concentration. The adsorption of Tl(I) was reduced by increase of temperature. Differences between adsorption isotherm become smaller at higher temperature. Langmuir and Freundlich adsorption models were used to fit the experimental data of Tl(I).

The equation for the Langmuir adsorption model is

$$Ce/Qe = Ce/Q\text{max} + 1/(Q\text{max} \times b) \tag{6}$$

where $Q_e$ is the adsorption amount (mg/g); $Q_{max}$ is the maximum adsorption amount (mg/g); $C_e$ is Tl concentration at adsorption equilibrium (mg/L); and b is the adsorption equilibrium constant (L/mg).

The equation for the Freundlich adsorption model is

$$\ln Qe = \ln k + (1/n)\ln Ce \tag{7}$$

Where $Q_e$ is the adsorption amount (mg/g); $k$ and $n$ are constants; and $C_e$ is Tl concentration at adsorption equilibrium (mg/L);

## Adsorption thermodynamics

The thermodynamic parameters provide in-depth information about the inherent energetic changes associated with adsorption. The standard enthalpy change ($\Delta H^o$), standard entropy change ($\Delta S^o$), and Gibbs free energy change ($\Delta G^o$) can be calculated from the

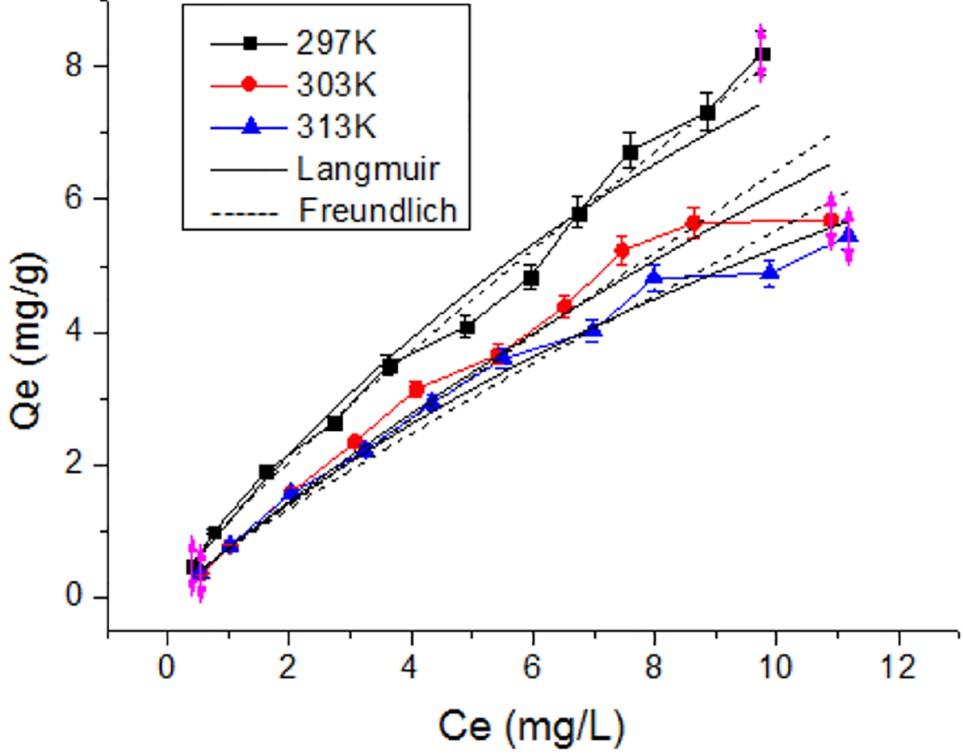

**Figure 6  Adsorption isotherms of Tl on rutile nano-TiO₂.** The adsorption amount of Tl(I) decreased with increasing temperature, and the a dsorption of Tl was exothermic.

temperature-dependent adsorption data using the following equations:

$$\Delta G = RT \ln Ks \tag{8}$$

$$\Delta G = \Delta H - T\Delta S \tag{9}$$

$$\ln Ks = -\frac{\Delta H}{RT} + \frac{\Delta S}{R} \tag{10}$$

where $R$ is the gas constant; $K_s$ is the equilibrium constant; and $T$ is the absolute temperature.

$\Delta H^o$ and $\Delta S^o$ were obtained from the slope and intercept of the linear plot of $\ln K_s$ versus $1/T$, respectively. The thermodynamic parameters calculated from Eqs. (5) to (7) are summarized in Table 2. The results showed that the adsorption amount of Tl(I) decreased with increasing temperature. The negative values of $\Delta H^o$ indicated that the adsorption of Tl was exothermic. This can be explained by the fact that the heat of adsorption of Tl(I) on RNTD exceeds the dehydration energy of the Tl(I) ions during adsorption. The negative values of $\Delta S^o$ showed that adsorption of Tl decreased the randomness of the liquid–solid system. The negative values of $\Delta G^o$ suggested that the adsorption of Tl was spontaneous.

**Table 2   Isotherm parameters for adsorption of Tl(I) on rutile nano-TiO$_2$.**

| T/K | $\Delta H$/(kJ mol$^{-1}$) | $\Delta S$/(kJ mol$^{-1}$K$^{-1}$) | $\Delta G$/(kJ mol$^{-1}$) |
|---|---|---|---|
| 298 | −2.19 | −60.67 | 0.4567 |
| 303 | −2.19 | −60.67 | −0.6763 |
| 313 | −2.19 | −60.67 | −0.5992 |

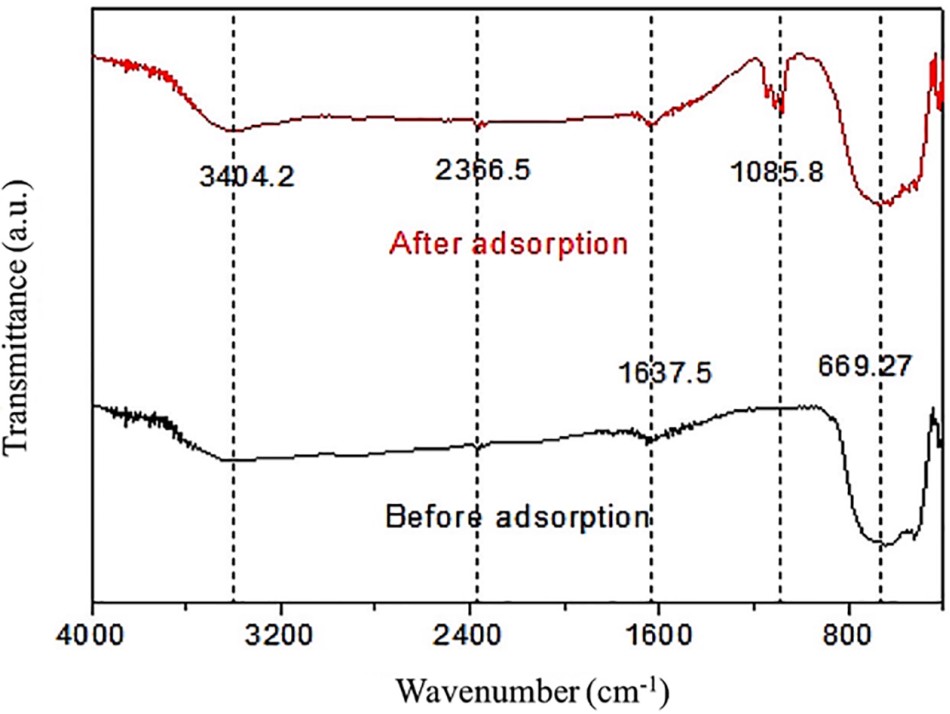

**Figure 7   Infrared spectra of rutile nano-TiO$_2$.** The oxygen-containing functional groups on RNTD were the main adsorption sites for Tl.

# CHARACTERIZATION OF TL ADSORPTION

## FTIR analysis

Figure 7 exhibited the FTIR spectra of pure RNTD and RNTD after Tl adsorption. The peaks at ~1,045 cm$^{-1}$ and the strong signal between 2,600~3,600 cm$^{-1}$ were ascribed to hydroxyl bond O–H stretching (water adsorbed on TiO$_2$) (*Tanzifi et al., 2018*). The wide peak between 600 and 800 cm$^{-1}$ corresponded to Ti–O–Ti bonds was observed at both of the spectra. The weak absorption peak at ~2,360 cm$^{-1}$ corresponds to the stretching vibration of TiOO-H. A strong peak attributed to ClO$_4^-$ is seen at 1,085.8 cm$^{-1}$. It can be due to the ion exchange between ClO$_4^-$ on the surface of the RNTD after addition of NaClO$_4$. A vibration peak due to physically adsorbed water appears at 1,637.5 cm$^{-1}$ (*Yousefzadeh, Salarian & Kalal, 2018*).
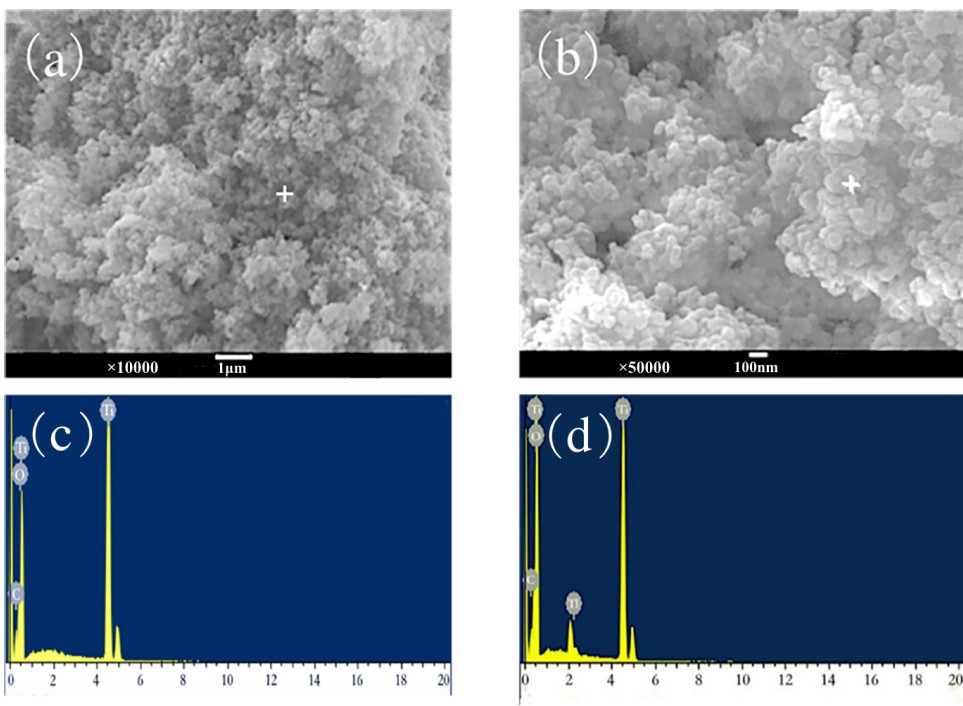

**Figure 8** SEM-EDS image after adsorption of nano-TiO₂ (A and C) before adsorption; (B and D) after adsorption.

## SEM and XRD analyses

As shown in the SEM images (Fig. 8), the surface of $TiO_2$ nano-particles before adsorption was covered by loose particles with diameter in nm, which implies a large specific surface area. After adsorption, due to the hydrophilicity of $TiO_2$, ultrafine nanoparticles moved randomly in the aqueous phase, diffused, and spontaneously agglomerated to produce secondary particle clumping. Thus, a dispersed system of micron-sized and nano-sized $TiO_2$ coexists in the solution, between the colloidal system and the coarsely dispersed system (Fig. 8).

The XRD patterns of pure RNTD and RNTD after Tl adsorption were presented in Fig. 9. Not obvious differences were observed between the two patterns. Anatase and rutile phases of $TiO_2$ were observed in both of the patterns at the $2\theta$ angles of 27.48, 36.12, 39.24, 41.28, 44.08, 54.36, 56.68, 62.80, 64.12, 69.04, 69.80°. It matched well with the standard data for $TiO_2$ diffraction pattern (JCPDS89-4920) (*Maleki et al., 2016*).

## ENVIRONMENTAL IMPLICATIONS

As shown in the study, RNTD has a fast adsorption rate and excellent adsorption amount for Tl(I). It is possible that Tl(I) would release from RNTD when ingested by organisms, which may greatly increases the risk of Tl to the environment and organisms. In addition, RNTD may be modified differently in the environment, complicating the interaction between Tl(I) and RNTD. Moreover, the prevalence of organic matter and other heavy

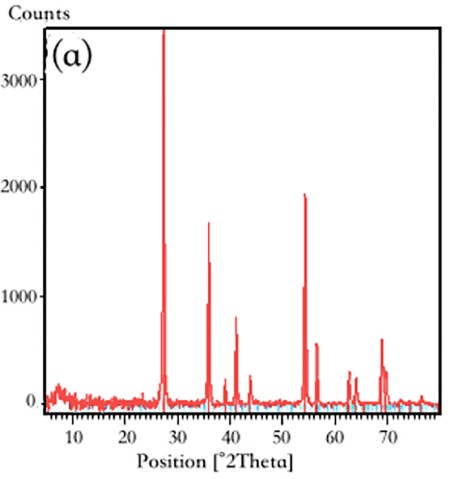 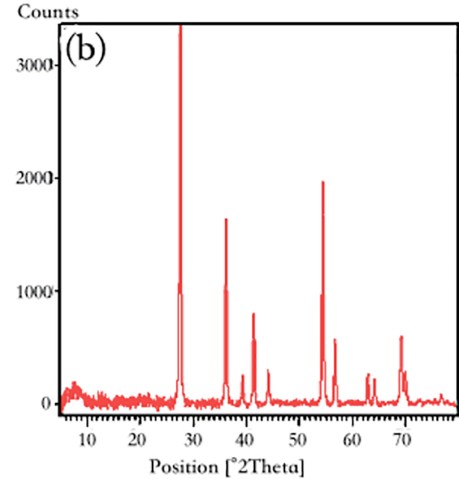

**Figure 9  X-ray diffraction pattern of nano-TiO$_2$ (A) before adsorption; (B) after adsorption.**

metal ions in aqueous environments may also contribute to the dispersion and adsorption of Tl(I) on RNTD.

## CONCLUSION

The results showed that the Tl(I) adsorption amount of nano-TiO$_2$ increased with increasing pH. Efficient adsorption of Tl(I) was found to occur even at a pH as low as 2, whereas the optimum pH for Tl(I) adsorption was approximately 9–10. The adsorption was rapid, with high removal efficiency of Tl (I) more than 40% in the first 15 min. The adsorption of Tl(I) on nano-TiO$_2$ fitted well to the Langmuir isotherm, with a calculated maximum adsorption amount of 51.2 mg/g at room temperature, indicating monolayer adsorption sites for Tl on the adsorbent surface. The adsoprtion was found to be an exothermic process. The pseudo-second-order equation could best fit the kinetics of Tl adsorption ($R^2 = 0.9969$). All these results suggest the pivotal role of RNTD on altering the transport, bioavailability and fate of Tl(I) in aqueous environment. More works are necessary to improve the understanding of the transport and fate of nano-titanium dioxide and Tl(I).

### Funding
This work was supported by the Natural Science Foundation of China (Nos. 41573008, 41873015, 41573119, 41773011 and U1612442), the Guangdong Provincial Natural Science Foundation (2017A030313247), the Environmental Protection Ministry of Public Welfare Research Projects (201509051), the Guangzhou University's 2017 training program for young top-notch personnel (BJ201709) and the 16th "Challenge Cup" Undergraduate Program and Provincial Undergraduate Training Project for Innovation (201811078128).

The funders had no role in study design, data collection and analysis, decision to publish, or preparation of the manuscript.

### Grant Disclosures

The following grant information was disclosed by the authors:

Natural Science Foundation of China: 41573008, 41873015, 41573119, 41773011, U1612442.

Guangdong Provincial Natural Science Foundation: 2017A030313247.

Environmental Protection Ministry of Public Welfare Research Projects: 201509051.

Guangzhou University: BJ201709.

16th "Challenge Cup" Undergraduate Program and Provincial Undergraduate Training Project for Innovation: 201811078128.

### Competing Interests

The authors declare there are no competing interests.

### Author Contributions

- Weilong Zhang and Yang Wu performed the experiments, analyzed the data, prepared figures and/or tables.
- Jin Wang conceived and designed the experiments, analyzed the data, contributed reagents/materials/analysis tools, approved the final draft, modify the manuscript.
- Juan Liu conceived and designed the experiments, analyzed the data, contributed reagents/materials/analysis tools, prepared figures and/or tables, authored or reviewed drafts of the paper.
- Haifeng Lu performed the experiments, contributed reagents/materials/analysis tools.
- Shuijing Zhai and Yongheng Chen analyzed the data, contributed reagents/materials/-analysis tools.
- Qiaohui Zhong performed the experiments, analyzed the data, prepared figures and/or tables.
- Siyu Liu, Chunling Huang, Xiaoxiang Yu and Wenhui Zhang prepared figures and/or tables.
- Wanying Zhong performed the experiments, prepared figures and/or tables.

### Data Availability

The raw data was provided in a Supplemental File.

### Supplemental Information

Supplemental information for this article can be found online at http://dx.doi.org/10.7717/peerj.6820#supplemental-information.

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
