# Peer review of "Adsorption of thallium(I) on rutile nano-titanium dioxide and environmental implications"

_PeerJ, doi:10.7717/peerj.6820_

## Round 0.1 · original submission · Major Revisions

In addition to the general reviewer comments, both reviewers indicated that the scientific discussion on the data interpretation needs improvement; please focus your revisions on the results and discussion section. I also suggest that once you have made revisions to your manuscript, you check the grammar carefully, perhaps having the manuscript edited by a native English speaker.

·

Basic reporting

This is a well-written manuscript. The article structure, figs, tables were shared in a professional way. Literature and references were also provided.

Experimental design

NO Comment.

Validity of the findings

The topic is interesting. It would be better if the author could point out the significant contributions and the potential of this work on expanding upon the novel ideas. Also, it would be better if the author could discuss the applications of those finding.

Additional comments

This manuscript titled “Adsorption of thallium on rutile nano-titanium dioxide and environmental implications” by Dr. Liu et al is very interesting. It is mostly well written. However, some specific issues should be stated clearly. That would be better for the readers to understand the experiment design and results. Take the following into consideration.

1. Line 18. Suggest sentence reconstruction.
2. Line 55. To evaluate the adsorption behavior of TI on RNTD, various parameters were chosen to conduct the study. Why the study chose these parameters instead of others as tested? Do these parameters have any special characteristics associated with adsorption behavior or any other reasons?
3. Line 66. Why you chose 25 C for the shake flask study?
4. Line 77. Please change " For evaluating" to " To evaluate".
5. Line 115. Please add the citations. In addition, it would be better if the author could compare your results with other published data, which is related to ionic strength.
6. Line 147. Please list the reasons why you chose the pseudo-first-order model?

Reviewer 2 ·

Basic reporting

The structure of material & methods and results report is straight forward and clear.
Reference was included properly.

Experimental design

Experimental design is solid.
however, the scientific discussion on the data interpretation needs improvement.

Validity of the findings

Significant results discussion needed in this manuscript

Additional comments

Line 65: more detail samples prep, the number of treatment and treatment procedures etc. needed.
Line 76: after 300 min?
Line 102: figure 1 shows a decrease in adsorption amount with increased dosage. Also, 0.5-2.5 is not matching the figure either.
Line 113: Please double check the data, since the constant adsorption rate did not mean saturated. its still absorbing Ti without decreased rate.
Line 168: explain more about 297K, 303K, 313K etc.
Line 216: what sample was it?
Line 217: where is the figure 10?

---

## Round 0.2 · accepted · Accept

Thank you for your efforts in revising your manuscript according to the reviewer comments.

·

Basic reporting

The authors have adequately addressed my concerns in this revision.

Experimental design

The authors have adequately addressed my concerns in this revision.

Validity of the findings

The authors have adequately addressed my concerns in this revision.

Additional comments

The authors have adequately addressed my concerns in this revision. I recommend this paper for publication.

Reviewer 2 ·

Basic reporting

The article meets the standards and has improved from the last edit.

Experimental design

The article meets the standards and has improved from the last edit.

Validity of the findings

The article meets the standards and has improved from the last edit.

Additional comments

Dear Authors,

I reviewed the comments. You provided a relevant explanation as well as additional information requested by reviewers.

The content of the manuscript is ready to publish. However, the font of the reference is different. I would suggest you go through the format and grammar check one more round before publish.

Thanks,
Jiayi